# Salinity in *Jatropha curcas*: A Review of Physiological, Biochemical, and Molecular Factors Involved

Marcelo F. Pompelli *, Alfredo Jarma-Orozco and Luis Alfonso Rodríguez-Páez

Faculty of Agricultural Sciences, University of Cordoba, Monteria 230002, Colombia; jarma@fca.edu.co (A.J.-O.); larguez@fca.edu.co (L.A.R.-P.)
* Correspondence: marcelo@fca.edu.co; Tel.: +57-304-426-7319

**Abstract:** *Jatropha curcas* is a woody-shrub species of the Euphorbiaceae family that is widely distributed in tropical and subtropical areas. The great interest in its cultivation lies in the potential for achieving elevated yields of a high-quality oil. Another characteristic that makes *J. curcas* promising is its ability to produce green energy even in high-salinity soils. For a commercial cultivation to be considered effectively competent to withstand these conditions, it must produce enough to offset production costs. There is no doubt that *J. curcas* is considered promising, but numerous pilot projects for the commercial planting of *J. curcas* have failed worldwide, mainly due to a lack of reliable scientific knowledge about the species, its food security, and (mainly) its instability in commercial fruit production. The main goal of this review was to compile published results on tolerance/resistance or sensitivity to salt stress in *J. curcas*. Updating the knowledge on this theme may allow for researchers to trace strategies for future studies of stress physiology in this promising oil seed species.

**Keywords:** metabolic profile; molecular biology; NaCl; purging nut; salinity tolerance

## 1. Introduction

Nine out of ten of the world's most polluted cities are located in India [1]. The larger the population, the greater the demand for energy, which is commonly obtained from petroleum products or burning coal [2]. In this context, greenhouse gas emissions (GHGs) are the driving force for global climate change. Deforestation accounts for over 20% of the world's GHG emissions [3]. It is of utmost importance to restore deforested while recycling carbon [3]. Environmental, economic, and social issues have led to the recent trend of using renewable sources of energy worldwide [4], so biomass has increased in popularity and acquired a significant share of the global energy mix over a relatively short time. However, several biomass resources have triggered wide criticism for compromising food resources, agricultural lands, and fresh water to produce energy crops [5], but from environmental and socioeconomic perspectives, biomass is easily available, non-edible, and resilient to climatic variations, thus rendering is a potential biofuel capital for a cleaner and greener future [1]. It was recently projected that the 100.00 (Megatons; MT) of diesel consumed in 2019–2020 [1] could reach 143 and 191 MT in 2025 and 2030, respectively, based on an increase of 6% per annum. Jet fuel is a major product, with an annual production of 253.06 MT, followed by gasoline, fuel gas, and diesel [6]. According to Achten, Verchot [7], *Jatropha curcas* can absorb 30 tons of carbon per year, and besides substituting biofuels for fossil fuel, the seed shells of *J. curcas* have a high energy value (~18–19 MJ kg$^{-1}$). Both husks and shells are not suitable as substrates in biogas digesters due to low digestibility [8]. Unblended 100% *J. curcas* methyl esters were extensively tested on the road in India with modern Mercedes cars. A total of 80,000 liters were used in these tests, and the overall results were highly satisfactory [8].

*J. curcas* has emerged as one of the most promising biomass feedstocks for jet biofuel (JBF) production, since it is non-edible and able to grow in non-arable lands with minimal water and energy requirements [6]. However, oil produced by *J. curcas* seeds remains distant

and noncompetitive with conventional Jet-A fuel available in the market of comparable price. Klein-Marcuschamer, Turner [9] investigated the potential of microalgae, Pongamia, and sugarcane in JBF production, and they found minimum selling prices of 2.42, 1.60, and 1.06 $/kg, respectively. Furthermore, Wang [10] reported a minimum JBF selling price of 1.14 $/kg. Moreover, Tao, Milbrandt [11] studied four oil-bearing crops including *Jatropha*, camelina, pennycress, and castor, among which *Jatropha* achieved the lowest production cost with a minimum JBF selling price of around 0.40 $ $kg^{-1}$—well below what IATA [12] presented in its 2021 bulletin (~0.60–0.64 $/kg).

Since the passing of related legislation [13], Brazil has seen a consistent increase in the percent of biodiesel mixed with diesel sourced from fossil fuels, from 2% in 2008 to 20% in 2025 [14]. Nevertheless, the main raw material used for biodiesel production in Brazil is soybean oil (68% in 2019), followed by animal fat (11%) [15]. However, soybean is a versatile crop that is especially important for human and animal nutrition, and *J. curcas* does not compete with the food market and could be an option to be explored. Considering that *J. curcas* has a productive life of approximately 50 years [16] and that it attains maximum productivity after 4–5 years, the necessary level of production is achievable [7]. Moreover, this species has shown the ability adapt to different agroclimatic conditions [17,18], including infertile soils, making it suitable for cultivation on degraded soils while protecting them from erosion [19]. Such characteristics make the cultivation of *J. curcas* suitable for familiar agriculture in marginal lands, where it could improve the standard of living and socio-economic status of farmers, particularly in developing countries [20].

The production of seeds in *J. curcas* starts around 6 months after planting [21], and they achieve maximum annual yields of 9–39 tons of seeds $ha^{-1}$ at 4 years after planting [22]. At the best stage of physiological maturity of the fruits, the seeds are expected to contain 36% moisture (humidity basis) [23] and up to 45% oil in the kernel [24]. Thus, it is possible to estimate the potential productivity of 25 tons of dry seeds $ha^{-1}$ (64% of 39 tons), and 5.2 tons oil $ha^{-1}$ $year^{-1}$ (45% of 15.4 tons when assuming 75% of extraction efficiency) or 5600 L oil $ha^{-1}$ $year^{-1}$ (assuming a density of 0.92 kg $L^{-1}$) [25]. Despite recent results supporting the idea that the productivity of seeds declines six years after planting [22], there have been other estimates of a 50-year productive life of *J. curcas*, thus strengthening the great production potential of the species under ideal conditions. To put it in perspective, soybean—the most cultivated oilseed crop in world, food purposes included—produces 675 L oil $ha^{-1}$ on average a year [26].

In April 2013, SG Biofuels (a privately held bioenergy crop company that grows and researches *J. curcas* for the production of biodiesel, biojet fuel, and specialty chemicals) signed up for more than 101,150 hectares in various field trial and deployment agreements—including an agreement for the trial of *J. curcas* with Bharat Petroleum in India with 34,803 ha for the first phase commercial deployment following the trials. A similar 30,350 ha deal was set up in Brazil with a consortium including JETBIO, Airbus, the Inter-American Development Bank, Bioventures Brazil, Air BP, and TAM Airlines. Moreover, during the last decade, numerous *Jatropha* projects have been implemented in Asia, Africa, and Latin America. The National Research Institute for Forestry, Agricultural and Livestock estimated that the potential lands for *Jatropha* cultivation in Mexico comprised more than 2.6 million hectares. Given this information, in 2007, various southeastern states in Mexico planned to cultivate thousands of hectares (ha), including Veracruz (200,000), Yucatán (6000), Michoacán (6000), and Chiapas (20,000) [27]. The goals of these projects have been diverse, but promoting sustainable rural development, reducing energy dependency, and GHG emissions have been frequent aims. However, doubts have recently been cast on the profitability of *J. curcas* and the financial viability of its cultivation [27,28].

The present review is a compilation of recent results regarding the effects of salt stress on the physiological, biochemical, and molecular aspects of the energy species *J. curcas*. Our main objectives were to provide updates on the state of of the issue and to propose new directions for future research on the stress physiology of *J. curcas*. Why has a plant species with tremendous apparent capacity to generate biodiesel and the possibility of being a

world leader in the achievement of green sustainability been the target of so much criticism? Why have many projects worldwide been abandoned before completion? What are the agronomic, physiological, and cellular challenges that we must address for the cultivation of the species? Which strategies are worth investigating and how can molecular biology help to understand the studied processes? This review is intended to clearly present these and other ideas, as well as to unify the extensive published data on the species.

## 2. Physiological, Biochemical, and Molecular Responses of *Jatropha curcas* L. to Abiotic Stress

Plants subjected to water deficits usually suffer from both 'thirst' (decreased water uptake and transport) and 'starvation' (decreased carbon and nutrient uptake). Indeed, due to the fast and efficient stomatal control of transpiration, rather than thirst, starvation is the main cause of drought stress-induced reductions in growth and biomass accumulation in *J. curcas*. Investigations into the influence of salt-stress deficits on the physiology of *J. curcas* have reported consistent salt-induced stomatal closure, followed by decreases in transpiration ($E$) and net photosynthesis ($P_N$) [29–33].

The duration of salt stress determines different plant responses in terms of the velocity and severity of the effects on leaf gas exchange variables and the velocity and degree of recovery after stress release. That was clearly demonstrated by Corte-Real, Miranda [21], who observed the fast and complete or slow and incomplete recovery of leaf gas exchange variables after a cycle of rapid salt-stressor and slow water-deficit treatments. The authors pointed out that rapid salt stress-induced stomatal closure, which kept the leaf water status almost stable, was responsible for the quickly recovery of leaf gas exchange variables when the plants were removed from the saline solution [21].

Osmotic adjustment (OA) is a physiological phenomenon observed in some plant species facing osmotic stress caused by salinity or water deficit [34]. OA consists of the active accumulation of organic and inorganic solutes in plant cells (Figure 1), which helps to attract water into the cell, thus maintaining positive turgor pressure [35]. Several types of solutes, such as amino acids, proline, glycine betaine, sugars (e.g., sorbitol, pinitol, and glycerol), and phenolic compounds accumulate in OA under osmotic-limited conditions.

Many authors [32,36–38] have described proline as an osmoregulator and as being promoted by water and salt stress. However, more recent studies [38–41] have shown that proline, as a key role in plants under drought and salt stresses, not only acts as an osmoprotectant but also functions as an antioxidant. In fact, the synthesis of proline comes from an alternative route of metabolizing glutamate, which is phosphated (by 1 mole of ATP) and reduced to proline from the consumption of 2 moles of NADPH [42]. As a consequence, a stressed plant tends to proportionally reduce the stomatal opening, which limits the entry of $CO_2$ into the system [29,43]. With less $CO_2$ in the system, the Calvin–Benson cycle is compromised and the ATPs and NADPHs that were formed and continue to form through chloroplastidic electron chain transport (ECT) remain in their reduced forms [44–46]. Assuming that ATP and NADPH remain in their reduced forms, there would theoretically be no oxidized forms to receive more electrons from ECT; in this case, the electrons would be quickly captured by $O_2$, forming the terrible reactive oxygen species (ROS) [47]. It is in this pathway that proline acts because, as previously reported for the formation of 1 mole of proline, 1 mole of ATP and 2 moles of NADPH are spent and the synthesis of proline enters the system as a stress preventer, not as an osmoregulator. The role of proline is similar to the role of photorespiration, which, in the absence of $CO_2$, can promote the re-oxidation of chloroplast ATPs and NADPHs and (in the same way as explained above) can serve as an escape valve that allows mesophyll cells to continue to promote photosynthesis even with stomatal opening [48] and its consequent compromised $CO_2$ entry.

In addition to proline and soluble sugars, such as pinitol and sorbitol, glycerol can also contribute to osmotic adjustment in plants [49]. The accumulation of potassium ($K^+$) is an effective contribution of the ion to osmotic adjustment. Nevertheless, genotype variation

in OA and in osmolyte accumulation has been documented for both *R. communis* [50] and *J. curcas* [51], which may be useful in breeding programs.

Tominaga, Inafuku [52] revealed that *J. curcas* is able to preserve the integrity of the photosystem (PS) II when the stomata are partially [29,32] closed in dry conditions, thus regulating the thermal dissipation and adjusting the quantum efficiency of the PSII to the $CO_2$ fixation capacity. ROS production during abiotic stress is the main cause of protein oxidation and lipid peroxidation, generating toxic molecules, including malondialdehyde (MDA) [18,53] and substances reactive to thiobarbituric acid (TBARS) [54–57]. In *J. curcas* leaves, the content of MDA was found to be 1.85-fold higher in 3.5 dS $m^{-1}$ of electrical conductivity compared to a control; in consequence, the electrolyte leakage was found to be 1.29-times higher in the same treatment [32].

Water-stressed plants have developed other mechanisms, including antioxidant defense systems, with enzymes such as superoxide dismutase (SOD; EC 1.15.1.1), peroxidase (POD; EC 1.11.1.7), and catalase (CAT; EC 1.11.1.6). While SOD acts on reduced oxygen derivatives (e.g., superoxide radical, $O_2^{\bullet-}$, singlet oxygen, and $^1O_2$ and hydroxyl radical HO) to form $H_2O_2$, CAT and APX enzymes act in the dismutation of $H_2O_2$ into $H_2O$ and $O_2$. In addition, plants can use non-enzymatic antioxidants consisting of low molecular-weight metabolites such as carotenoids, ascorbic acid, glutathione, tocopherols, and phenolic compounds [58], which increases their ability to grow and develop in significant salinity conditions. The balance between ROS production and antioxidant activity will determine different responses in plant cells, such as signaling and oxidative stress [59]. Increases in SOD, CAT, and ascorbate peroxidase (APX) activities would demonstrate the strong participation of the enzyme system in *J. curcas* induced by water deficits for 4, 8, and 18 days, in addition to aiding the process of recovery from drought (Pompelli et al., 2010). In turn, the maintenance of redox homeostasis can contribute to plant drought tolerance [60].

## 3. How Does Salinity Modulate Physiological, Biochemical and Molecular Responses in *J. curcas*?

As soil salinity increases, the extraction of water from the soil becomes increasingly difficult for plants [61]. Surprisingly, an estimated one to two percent reduction in the world's arable area is observed yearly as an effect of salinization, indicating the severity and importance of secondary salinization. In addition, salinity has often been associated with the decreased yield and stability of some crop species [62].

### 3.1. Advantages and Disadvantages of Wastewater in Agriculture

In the global context, it is evident that recycled urban wastewater (RWW) in agriculture will increase in water-scarce countries in the near future, particularly in the vicinity of cities [63,64]. Irrigation with RWW has been used for three purposes: (*i*) as a complementary treatment method for wastewater; (*ii*) as an available water source for agriculture—a sector demanding ~70% of the consumptive water use in Brazil; and (*iii*) as a nutrient source associated with mineral fertilizer savings and high crop yields [16,65]. For this role, Brazil deserves special attention, since a record export of agricultural products such as soybeans, corn, cellulose, sugarcane, and coffee that should exceed 80.87 billion tons is expected for 2022 [66]. Thus, if Brazil could replace 10% of freshwater with wastewater in agriculture, a few billion $mm^3$ of freshwater would be spared, thus making Brazil the focus of attention for the gradual use of RWW instead of freshwater, not for its ecological disasters [64]. Various studies have shown that the soil application of RWW as a water and nutrient source for agricultural irrigation represents a low-cost alternative [67] and is applicable in both dry and humid regions. Hence, irrigation with RWW has practical importance regarding nutrient availability, especially in the acidic soils of Brazil that have inherently low natural fertility [65]. In Morocco, 700 mil $m^3$ of RWW was discharged without treatment, an amount that is predicted to increase to 900 mil $m^3$ in 2020 [63], and just about 70 mil $m^3$ are being re-used each year. The benefits of RWW use additionally include the conservation of fresh water reserves, reductions in the discharge of pollutants into water bodies, potential

for the recycling of both water and nutrients that otherwise would have been disposed of into the environment and consequently contaminate natural water bodies, and the supply of nutrients to crops due to the organic matter and nutrients present in this water [68,69].

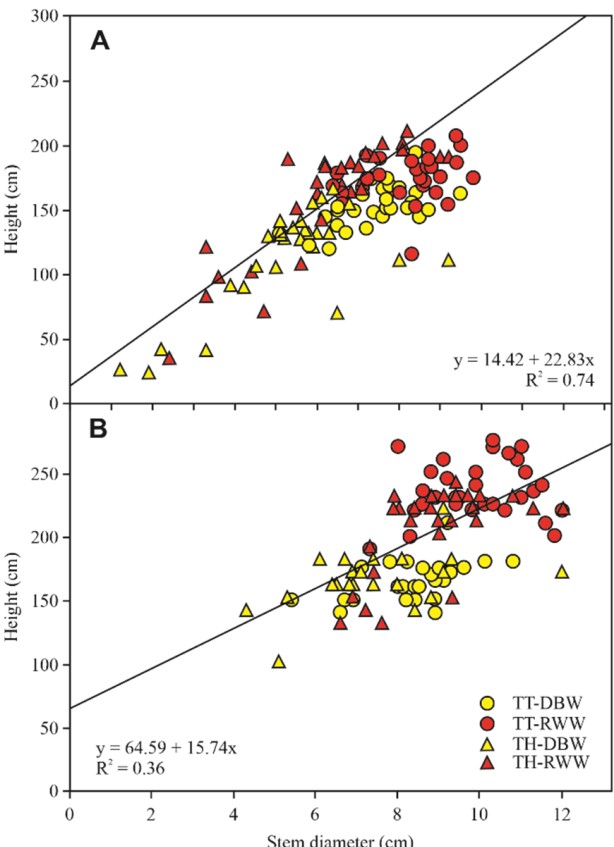

**Figure 1.** Stem diameter (**A**) and height (**B**) of *Jatropha curcas* at two different stages of cultivation and under different treatments. Figure adapted from [68].

Clay-loam soils, generally with higher concentrations of nutrients than sandy soils, may be more suitable for *J. curcas* cultivation [70]. Many scholars have also noted the preference of *J. curcas* for sandy soils, which are more favorable to root development than clay soils [16,65]. Dorta-Santos, Tejedor [68] cultivated *J. curcas* plants in two soil types, sandy-loam soil (TT) and clay-loam soil (TH), plus two irrigation forms, RWW and desalinated brackish groundwater (DBW) (Figure 1). These authors showed that relationship between the base diameter and height of the plants indicated that both morphometrical variables were significantly affected by the soil and water factors but not by the interaction between them. Thus, for the same soil type, irrigation using RWW proved better for plant growth than DBW irrigation, and for the same type of water, the TT proved more favorable than TH. Regarding seed production, these authors reported that production under RWW irrigation increased 12-fold with respect to that obtained using DBW (mean ~2258 versus 197 kg·ha$^{-1}$). Additionally, due to the plant's relatively wild nature and the enormous gaps in knowledge about its genetic background and optimized agronomic practices, many *J. curcas* projects have failed to reach commercially viable yields [28,71] and were subsequently abandoned.

### 3.2. Sodium (Na$^+$) and Chloride (Cl$^-$) Compartmentalization Related to Salt Stress Tolerance

Concerning appropriate soil characteristics for growing *J. curcas*, the data published by different authors have often proven contradictory. Some authors have considered *J. curcas* to be a salt-sensitive plant [21,37,63,72], while others have considered *J. curcas* to be moderately sensitive [73] or moderately tolerant to salinity [63,68,74–77]. Accordingly, the potential

of *J. curcas* in terms of tolerance to salinity remains controversial. This divergence may be related to the diversity of physiological responses among genotypes in different studies and a lack of high yielding variety. An understanding of *J. curcas*'s response mechanisms under salt stress would thus provide valuable information for improving stress tolerance. It seems plausible that understanding the physiological and molecular mechanisms underlying salt stress resistance (SSR) is a promising and cost-effective strategy for developing highly resistant cultivars.

The adverse effects of salinity are the results of three primary factors: first, the inhibition of water uptake caused by the low water potential of the soil; second, the toxic effects of the $Na^+$ and $Cl^-$ ions at the cellular level; and third, the alteration of the nutritional balance, resulting in high ratios of $Na^+/Ca^{2+}$, $Na^+/K^+$, $Na^+/Mg^{2+}$, $Cl^-/NO_3^-$, and $Cl^-/H_2PO_4^-$. Plants have developed complex mechanisms for the avoidance of salinity stress. These mechanisms include (*i*) the accumulation of significant amounts of ions to ensure sufficient water uptake (a process known as osmotic adjustment; Figure 2A), (*ii*) the ability to avoid ion translocation from the root to the aerial part, (*iii*) the compartmentalization of toxic ions within the cell, and (*iv*) a mechanism of $Na^+$ exclusion via export across the plasma membranes [35,78–80]. In seedlings, increasing salt levels were found to result in a consistent decrease in total water loss associated with an increase in $Na^+$ concentration in leaves, while in 3-year-old *J. curcas*, the total water loss of plants treated with 25 mmol of NaCl was higher than that of control plants and then decreased at higher NaCl levels until 100 mmol. The total water loss values at high NaCl treatments (100, 150, and 200 mmol of NaCl) were lowest and of similar magnitude [63].

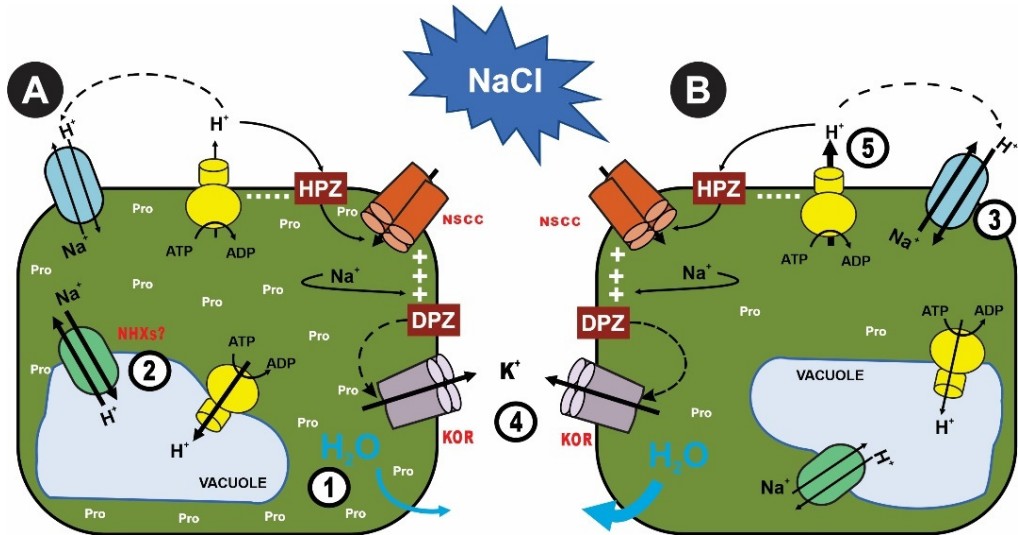

**Figure 2.** (**A**) Osmotic adjustment through the accumulation of compatible solutes such as proline in leaves and roots to prevent the loss of water (1). Na+ compartmentalization in vacuole-like structures suggests the involvement of vacuolar and endosomal $Na^+/H^+$ antiporters (NHX) (2). (3) $Na^+$ extrusion or efflux from the cytosol to the apoplast by SOS1 exchangers (3). (**B**) Regulation of $K^+$ homeostasis through $K^+$ retention in roots is crucial in plants (4). (5) Increase in the $H^+$-ATPase activity to cell membrane repolarization or $Na^+$ extrusion. KOR: outward-rectifying $K^+$ channels (5). HPZ is hyperpolarization, and DPZ is depolarization. Modified from the work of Quintal, Velarde-Buerdía et al. (2014).

The progressive accumulation of salt in photosynthetically active mesophyll tissues leads to the further inhibition of $CO_2$ assimilation because it affects the processes of photosynthesis in chloroplasts. This involves suppressing the activity of photosynthetic enzymes of the Calvin cycle, disturbing chlorophyll biosynthesis, and reducing the efficiency of the operation and structural integrity of the photosynthetic apparatus and thylakoid membranes (revised by Pan, Liu [81]). These authors presented a table describing some key

photosynthetic enzymes and structural proteins, including a substantial increase in cytosolic and stromal $Na^+$ concentrations that inhibit many key metabolic processes such as protein synthesis and oxidative phosphorylation. Increased cytosolic $Na^+$ levels also disturb intracellular $K^+$ and $Ca^{+2}$ homeostasis in leaf mesophylls, thus affecting photosynthesis and light signal transduction [81].

How to improve plant salt stress tolerance has become a focus of plant biology research. Some authors have reported that $Cl^-$ uptake and transport in plants is less controlled than those of $Na^+$ [80]. The result is a wide range of physiological and biochemical changes leading to the inhibition of growth and development; reductions in photosynthesis, respiration, and protein synthesis; and the disruption of nucleic acid metabolism [82]. To survive salt stress, plants respond and adapt through sophisticated mechanisms. For this, modifications of membrane transport, signal transduction, redox reaction, and other processes have been shown to be involved in the salt-stress response [36].

Corte-Real, Miranda [21] studied the effect of NaCl on growth and gas exchange in *J. curcas*, and they reported that these parameters were more seriously affected in salt-treated plants, though this effect were dose- and genotype-dependent. However, even the plants that were the most affected by NaCl could be restored after 6 days in recovery conditions, suggesting that *J. curcas* is moderately sensitive to salt. This finding was in agreement with that of Cabral, Binneck [74], who first expressed genes after a short salt stimulus and showed that the expression of a protoplast inner protein in *J. curcas* (*JcPIP1* and *JcPIP2*) was significantly decreased in the roots of *J. curcas* plants treated with 150 mM of NaCl. In *J. curcas*, salt tolerance was achieved by silencing of the *JcPIP1* and *JcPIP2* using tobacco rattle mosaic virus (TRV)-based on virus-induced gene silencing (VIGS). At 3 days after infiltration with *TRV:JcPIP1* or *TRV:JcPIP2*, the leaves of *J. curcas* plants became yellowish, and 10–18 days later, the leaves of the treated plants displayed a withering phenotype [83]. Plants infiltrated with *TRV:JcPIP1* or *TRV:JcPIP2* displayed necrosis spots on the leaves under a salt stress condition; Azevedo, Santos-Rosa [84] described upregulated roots, the plasma-membrane-intrinsic protein PIP2;2, and four tons of plastic. These authors also described the putative 1-aminocyclopropane-1-carboxylate oxidase 4 and S-adenosylmethionine synthetase (SAM2) associated with the ethylene biosynthesis pathway.

### 3.3. Effect of Salt on Cellular Metabolism and Defense

Among all salt-tolerance mechanisms, the exclusion of $Na^+$ and $Cl^-$ from the cytosol via compartmentalization into vacuoles has been most frequently observed in glycophyte species [78,79,85]. The overaccumulation of excessive $Na^+$ in the plant cytoplasm can be prevented using three main strategies in plants: (*i*) inhibiting $Na^+$ influx, (*ii*) elevating vacuolar $Na^+$ compartmentation, and (*iii*) enhancing $Na^+$ efflux (Figure 3). The adaptive response to salinity is multigenic in nature; however, a single gene can also increase the salt tolerance of a plant species [86]. When plants suffer from high salinity, $Na^+$ can enter cells through non-selective cation channels (NSCCs), $K^+$ permeases, and other transporters. $Na^+/H^+$ antiporters are expressed by functional gene(s), play an important role in plant salt tolerance, and are examples of single genes that can increase salt tolerance [35,87]. Ion homeostasis is carried out either via the sequestration of excess sodium into vacuoles via vacuolar $Na^+/H^+$ antiporters (NHX1) energized by the proton gradient generated by vacuolar membrane $H^+$-ATPase and $H^+$-pyrophosphatases [87,88] or via active exclusion through $Na^+/H^+$ antiporters in salt overly sensitive (SOS) 1 located on the plasma membrane [86].

Jha, Mishra [87] expressed the *SbNHX1* from *Salicornia brachiata* in *J. curcas* and confirmed a salt tolerance of up to 200 mM of NaCl through the estimation of different growth parameters, such as increased sodium and chlorophyll levels, reduced rates of electrolyte leakage, and less malondialdehyde in the leaves, all indicative of reduced oxidative damage in these transgenic plants. $H^+$-ATPase also provides the driving force for potassium retention, which is also driven by sodium exclusion through NHX exchangers. The differential expression/function of SOS1 and HKT (high-affinity potassium transporters) depends

on habitat, demonstrating the complexity of salinity tolerance [89]. However, the development of transgenic *J. curcas* plants via the overexpression of selected salt-responsive gene(s) is a better choice for the genetic improvement of the plant regarding salt tolerance and breeding [87].

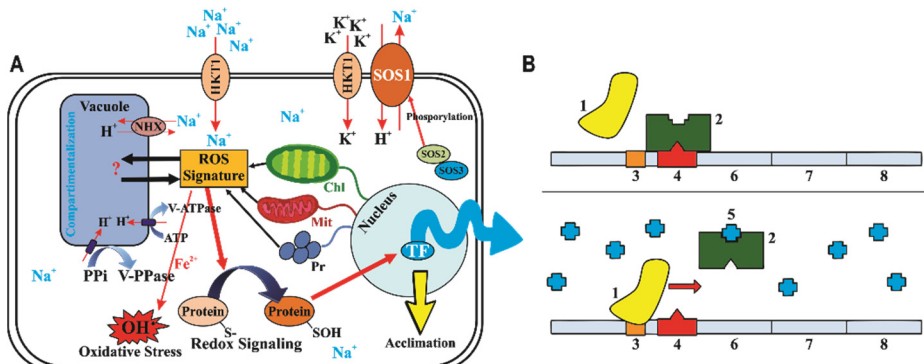

**Figure 3.** (**A**) The interaction between reactive oxygen species (ROS) produced from $Na^+$ signals. The ROS signature, promoted by an unknown pathway, compartmentalizes $Na^+$ into vacuoles. In the tonoplast, $H^+$ pumps equilibrate this compartmentalization. The ROS signature promotes the upregulation of a protein oxidation system that connects to the nucleus, promoting transcription factor (TF) activation and promoting some regulation of mitochondria (Mit), chloroplasts (Chl), and peroxisomes (Pr). In another way, the SOS1 channel in the plasma membrane promotes an antiporter flux of $Na^+$ and $H^+$. Na+ and $K^+$ compete with the same plasma membrane channel (HKT1). (**B**) Model proposed for the interaction between NaCl and catalase (CAT) gene expression. On non-salt-stressed plants (upper panel), the RNA polymerase (1) does not translate the CAT genes because there is a repressor (2) occupying the operon (4) on the CAT-encoding gene. Therefore, in salt-stressed plants (bottom panel), the $H_2O_2$ (5) binds to repressor that remains inactive and then dissociates from the CAT-encoding gene. Afterwards, the RNA polymerase binds to promoter region (3) on the CAT-encoding gene and then promotes the translation of CAT1 (6), CAT2 (7), and CAT3 (8). Modified from the work of Choudhary, Rivero [85] and Muchate, Nikalje [79].

Several halophyte genes associated with cation/proton antiporters on the plasma membrane (*SOS1*), vacuolar membrane (*NHX*), plasma membrane, vacuolar $H^+$-ATPases, vacuolar $H^+$-pyrophosphatase, and potassium transporters (as well as other genes involved in salt-tolerance mechanisms apart from defined pathways, i.e., crosstalk, osmolytes, antioxidant enzymes, salt glands, and bladders) are the current targets of researchers seeking strategies to transform glycophyte plants so that they can tolerate high salinity. Ubiquitous membrane proteins that catalyze the electroneutral exchange of $Na^+$ or $H^+$ across the membrane, thereby playing important roles in cellular $Na^+/K^+$ and pH homeostasis, are the most studied mechanisms to cope with salinity (Figure 2A) [79,85]. Among the five identified *SOS* genes, only three genes (*SOS1, SOS2,* and *SOS3*) are involved in the salt-signaling pathway for salt tolerance in *Arabidopsis* [38]. It is probable that vacuolar antiporters are more suitable and promising genes for genetic transformation because the entire process takes place in the cell itself, unlike *SOS1*, which has shown different functions depending on the external saline environment [79,85]. The removal and sequestration of $Na^+$ and the retention and uptake of $K^+$ from the cytoplasm are the mechanisms that plants use to reduce excess $Na^+$ or maintain $K^+$ content, respectively. In addition, the genes involved in the synthesis of compatible solutes or osmoprotectants such as proline, glycine-betaine, and inositol may comprise another interesting tolerance mechanism to protect cells from salinity damage [90].

The regulation of $Na^+$ plasma membrane signaling (HKT1) by calcium [79], phosphorylation, hormones such as nitric oxide (NO) [78] and calcium-dependent protein kinase (CDPKs) [79], or NADPH availability may generate signaling ROS at the apoplast. This signaling ROS then moves into the cytoplasm via regulated aquaporins, and together with

the metabolic and signaling ROS produced in the chloroplast, peroxisome, and mitochondria (Figure 2A), they alter the redox status of key regulatory proteins such as transcription factors (TFs) affecting gene expression, thus promoting the expression of specific genes such as *SOS2* and *SOS3* that activate via phosphorylation with *SOS1* (Figure 3A) to exchange $Na^+/H^+$ antiporters and CAT in the plasma membrane [79,85]. NO mediates the post-translational modification of target proteins through S-nitrosylation and nitration. Under water-deficit conditions, induced or not by salt stress, ABA induces NO and ROS synthesis. Both NO and ROS form 8-nitro-cGMP-inducing stomata closure that, in turn, reduces $CO_2$ assimilation [91]. S-nitrosylation, the covalent binding of NO to thiol groups of cysteine, is a post-translational modification that can regulate the function of some enzymes involved in respiration, antioxidation, and photorespiration during salinity stress [92]. Gadelha, Miranda [78] reported that NO-pretreated *J. curcas* seedlings presented a significant mitigation of salt stress due to low toxic ion and ROS accumulation via powerful antioxidant systems, resulting in the elevated growth of NO-treated seedlings under saline conditions (Figure 4). Moreover, salt stress was found to promote significant increases in lipid peroxidation and $H_2O_2$ contents at all analyzed time points and plant tissues; nevertheless, the NO pre-treatment effectively alleviated oxidative damage and ROS accumulation in *J. curcas* seedlings [78] (Figure 3). These findings suggest that the improved $K^+/Na^+$ of NO-treated stressed plants could be related to restrictions in $Na^+$ influx across the root plasma membrane or to vacuolar compartmentalization, contributing to enhanced salt tolerance [79]. Gadelha, Miranda [78] suggested that exogenous NO can also partially prevent lipid peroxidation and ROS accumulation, thereby alleviating the oxidative damage normally caused by salinity stress.

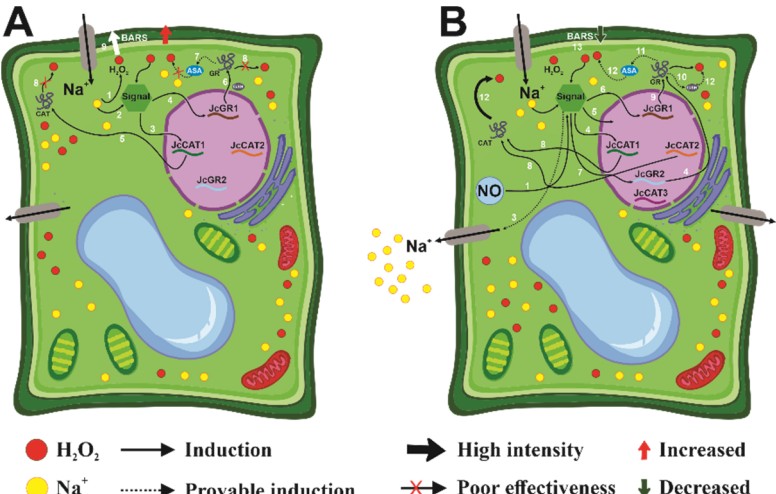

**Figure 4.** Model proposed for responses to salt stress. (**A**) Upon salt stress exposure, (1) the accumulation of sodium ($Na^+$—yellow circles) inside the cells promotes an increase in hydrogen peroxide ($H_2O_2$—red circles) content. In parallel, (2) a still-unknown signaling pathway recognizes the $H_2O_2$ and $Na^+$ accumulation and triggers an upregulation of (3) *JcCAT1* and (4) *JcGR1* gene expression, resulting in improved (5) catalase (CAT) and (6) glutathione reductase (GR) activities, respectively. The enhanced GR activity is caused by a greater (7) ascorbate (ASA) content. Nonetheless, these responses are not effective (8) for $H_2O_2$ scavenging, resulting in severe damage to (9) membrane lipids. (**B**) Yet, NO priming (1) quickly activates the signaling pathway, reprogramming the salt responses. First, (2) it recognizes $Na^+$ accumulation and seems to activate (3) the effective extrusion of $Na^+$ back into the medium. Secondly, it induces the upregulation of (4) *JcCAT1*, (5) *JcCAT2*, (6) *JcGR1*, and (7) *JcGR2* gene transcripts, which better direct (8) CAT and (9) GR activity. At the same time, the GR increases the contents of (10) glutathione (GSH) and (11) ASA. Finally, (12) the $H_2O_2$ is almost scavenged, (13) avoiding damage to membrane lipids in embryo axis cells and enhancing the salt tolerance of *J. curcas* seedlings. Modified from the work of Gadelha, Miranda [78].

Plants have several strategies to protect the photosystems against photoinhibition and photodamage, such as downregulation in light harvesting, excess energy dissipation by non-photochemical quenching, increases in water–water cycle and cyclic electron flow, and other important biochemical pathways such as the ascorbate–glutathione cycle, photorespiration, and nitrate and ammonia assimilation. The role of the efficiency of these processes in salt tolerance is possibly species-dependent, but the involved mechanisms are not understood [93]. The PSII D1 protein is the specific light-induced protease required to remove damaged D1 proteins, which are replaced with new copies produced by a de novo synthesis process termed reversible photoinhibition. At the molecular level, salt stress affects the repair via inhibiting the expression of the *psbA* genes for preD1 at both the transcriptional and the translational levels [94]. Salt stress also inhibits the degradation of the D1 protein in the photodamaged PSII. Similar results have also been reported for higher plants (reviewed in [81]). The involvement of the nitrate assimilation pathway is also closely linked to photosynthesis because nitrite and ammonia assimilation may consume electrons from reduced ferredoxin, thus reducing the overproduction of ROS, due to lower levels of $NADP^+$ [95]. The re-assimilation of $NH_4^+$, which arises from glycine decarboxylation, consumes reducing equivalents from ferredoxin and/or NAD(P)H, as well as ATP. Similarly, the conversion of glycerate to glycerate-3-phosphate (PGA) consumes ATP and regenerates adenosine diphosphate (ADP). The requirement of NADH for hydroxypyruvate reduction could provide an additional sink for reducing equivalents, which could originate from chloroplasts via the malate valve or the mitochondrial tricarboxylic acid (TCA) cycle. Thus, photorespiration is a significant and important component of the processes involved in minimizing ROS production (by dissipating excess reduced equivalents and energy), even though it is itself a source of $H_2O_2$ [96].

According to the work of Corte-Real, Oliveira [75], proteome data of salt-stressed *J. curcas* have evidenced of major molecular processes regulated by saline conditions that appear to be genotype-dependent (Figure 4). Differential proteomic analysis data revealed that the tolerant-like genotype showed proteins of different pathways related to the salinity response, including the production of antioxidant enzymes, as well as signaling and stress regulation pathways, especially with high sensitivity and responsivity to abscisic acid (ABA). The higher physiological efficiency of the tolerant-like genotype is also due to its ability to produce important enzymes from different energy and metabolic pathways, such as photosynthesis and glycolysis, thus ensuring their development. Though the sensitive-like genotype showed the exclusive accumulation of the glutarredoxin (Grxs) class protein, the tolerant-like genotype accumulated CAT, with both acting on $H_2O_2$ dismutation. Unlike CAT, Grxs are proteins of the glutathione cycle, an NADPH-dependent pathway. Thus, the reduction in NADPH production under salt stress may be part of this antioxidative pathway, suggesting that the sensitive-like genotype leads to the greater activation of a less efficient pathway in ROS scavengers compared to CAT activity in the tolerant-like *J. curcas* genotype [75].

### 3.4. Metabolite Profile of Salt-Stressed J. curcas Plants

In excess, ROS can result in extensive damage to proteins, DNA, and lipids, thereby impairing normal cellular functions and resulting in perpetual metabolic dysfunction and plant death [97]. It has been well-documented that plant species employing highly efficient antioxidant systems have improved salt stress tolerance, a response that may be signaled by some small molecules, such as $H_2O_2$, oligochitosan, NaCl, proline, ABA, and NO (see the work of Gadelha, Miranda [78] and references therein).

Lin, Li [98] reported that eight $CO_2$ assimilation-related proteins—two RuBisCO activases, RuBisCO large subunit-binding protein subunit beta, ribulose 1,5-bisphosphate carboxylase large subunit, transketolase, fructose 1,6-bisphosphatase, sedoheptulose 1,7-bisphosphatase, and ribose-5-phosphate isomerase—showed significant changes in response to salt treatment and were downregulated under salt stress (Figure 5). It was reported that salt stress suppressed the carboxylase activity of RuBisCO while enhancing

its oxygenase activity [98]. Moreover, in the energy category, a root mitochondrial ATP synthase beta subunit was found to be downregulated by salt stress, potentially causing low ATP production under high salt stress (Figure 6).

Downstream from this transcriptional regulatory network, a number of osmolyte (such as trehalose, raffinose, and proline) synthesis genes are involved in osmotic homeostasis and the stabilization of proteins and cellular structures [99]. Similarly to Zhang, Zhang [99], our team studied *J. curcas* subjected to 150 mM of NaCl for 3 h and detected 2.4-fold more raffinose in the salt-sensitive-like genotype and 12.7% less raffinose in the salt-tolerant-like genotype [77].

### 3.5. Transcriptome Analysis of Water and Salt-Stressed J. curcas Plants

A transcriptomic analysis via RNA sequencing in Illumina HiSeq™ 2000, performed for samples collected under moderate and severe stress followed by rewatering, revealed shoot- and root-specific molecular adaptations under moderate stress [100]. Under severe stress, a dramatic transcriptomic reorganization at the root and shoot level surpassed organ specificity. Roots respond rapidly to drought, with more than 400 reported differentially expressed genes (DEGs) under moderate stress. Souza, Silva [77] described 4646 suppressed DEGs in sensitive-like *J. curcas* and only 57 DEGS were in salt tolerant-like *J. curcas*. In leaves, however, impressive changes in expression patterns were only observed under severe stress, indicating that *J. curcas* leaves do not respond to drought as rapidly as roots, even though the leaf transcriptome returned to control conditions faster than the root transcriptome after stress relief [100].

Based on the existence of conserved Tryp, Arg, Lys, and Tyr motifs in comparison to the *Arabidopsis* genome, a total of 58 Tryp, Arg, Lys, and Tyr genes were identified in the *J. curcas* genome [101]. The analysis of transcript accumulation, based on the RNA-seq of these 58 Tryp, Arg, Lys, and Tyr genes, showed 26 transcripts with at least 2-fold increases or decreases in expression in response to drought and 17 transcripts that responded to drought and salinity. These genes are good candidates for improving tolerance to both stresses with a biotechnological approach.

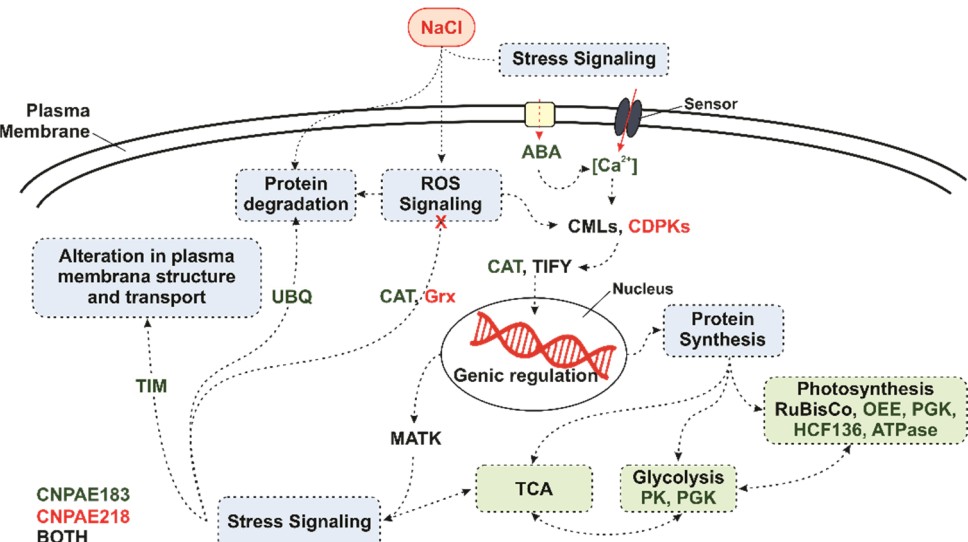

**Figure 5.** Model proposed for the gene modulation pathway in response to NaCl in two different genotypes of *Jatropha curcas* subjected to 48 h of 750 mM of NaCl. ABA, abscisic acid. CMLs, calcium-binding proteins. CDPKs, calcium-dependent protein kinase. MYB, MYB Transcription factor family. TIFY, TIFY transcription factor family. RuBisCO, ribulose 1,5-bisphosphatase carboxylase-oxygenase. OEE, oxygen-evolving protein. PGK, phosphoglycerate kinase. HCF136, photosystem II assembly/stability factor. ATPase, ATP synthase. PK, pyruvate kinase. MATK, maturase K. TIM, translocase of the mitochondrial inner membrane. UBQ, ubiquitin transferase. CAT, catalase. Grx, glutaredoxin. Modified from the work of Corte-Real, Oliveira [75].

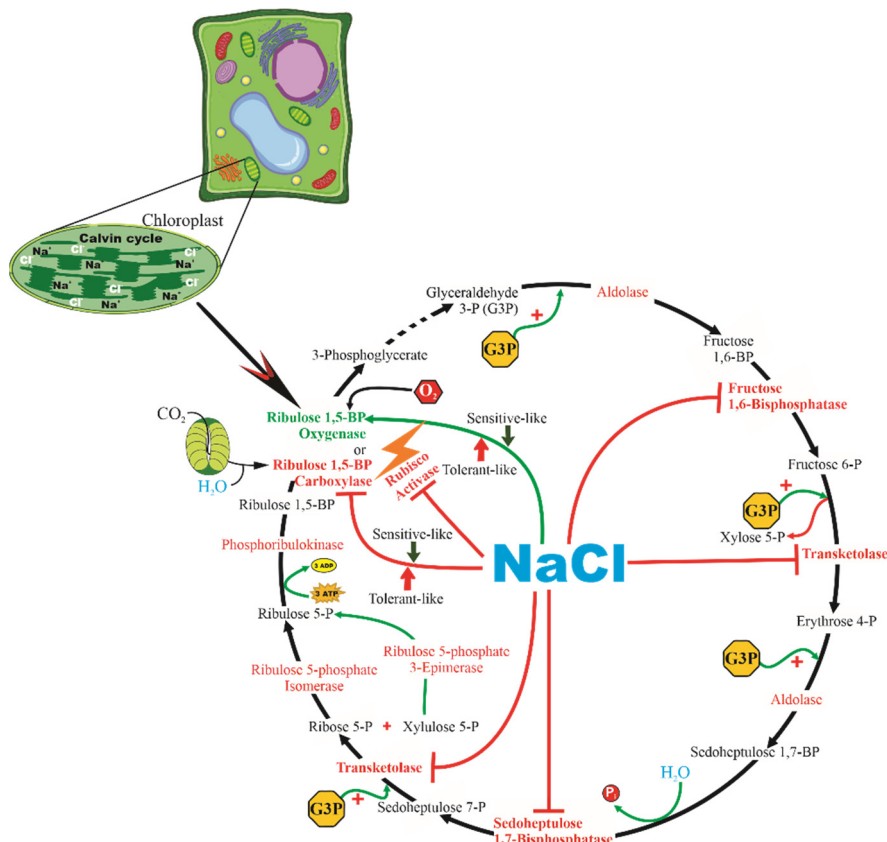

**Figure 6.** Model proposing that the Na$^+$ and Cl$^-$ ions promote the downregulation of many enzymes, such as RuBisCO activase, fructose 1,6-bisphosphatase, transketolase, sedoheptulose 1,7-bisphosphatase, and ribulose 5-phosphate isomerase, inhibiting the conversion of ribose 5-phosphate to ribulose 5-phosphate. On tolerant-like plants, Na$^+$ and Cl$^-$ promote the downregulation of the carboxylase activity and promote the oxygenase activity of ribulose 1,5-bisphosphatase carboxylase-oxygenase. On the other hand, on sensitive-like plants, Na$^+$ and Cl$^-$ promote the downregulation of the oxygenase activity of ribulose 1,5-bisphosphatase carboxylase-oxygenase, thus promoting carboxylase activity. Model constructed using data presented by Lin, Li [98] and Sivakumar, Sharmila [48].

Plant hormone signals, such as JA and ABA, play indispensable roles in plant response to drought stress [102]. Overexpressing downstream transcription factors (NAC, MYB, C$_2$H$_2$, CBF/DREB, WRKY, and bHLH) can also improve plant drought tolerance [103]. The C-repeat-binding factor/dehydration-responsive element-binding factor (CBF/DREB) is widely considered a key regulator of plant abiotic stress response [53]. Transgenic *N. benthamiana*, overexpressing *JcCBF2* of *J. curcas*, showed improved survival rate under drought stress [104]. Its leaf water retention and active oxygen-scavenging capacity but reduced photosynthesis and transpiration rate suggest that *JcCBF2* plays an important role in improving plant drought tolerance. Overexpressing *JcCBF2* was shown to lead to a decreased leaf area and increased leaf thickness. These overexpression lines of transgenic plants showed the expression of *NbMYB21*, *NbMYB86*, and *NbMYB44*, and both ABA- and JA-related genes increased under drought stress. These data indicate that *JcCBF2* is able to positively regulate the plant drought response by changing the leaf anatomical structure and possibly through JA and ABA signaling pathways.

In Supplementary Table S1, we show the gene(s) studied, the analytical methods applied, the plant conditions, the salinity treatments, the exposure time, and related references. In this set of manuscripts, *J. curcas* genes showed participation in phytohormone signaling, the synthesis of osmoprotectant compounds, redox metabolism, ionic and osmotic adjustment, and secondary metabolism. A general analysis of Supplementary Table S1 demonstrates that in addition to roots (the tissue in direct contact with NaCl), leaves were

also targeted. Additionally, most studies on *J. curcas* genes have concerned transgenic events caused by *Agrobacterium* in *Arabidopsis* or other model plants, followed by the validation of the transgene expression, usually by qPCR assay. In the studied set of manuscripts, the majority of *J. curcas* plants developed up to two months after germination when they received a salt treatment (which ranged from 50 to 300 mM of NaCl), with samples being collected at variable periods of hours (up to 48 h), days (from 2 to 25 days), or weeks (up to three weeks). These studies have contributed to *J. curcas* breeding programs by increasing salt tolerance in this species and other economically important and equally salt-sensitive ones.

After saline stress (100 mM of NaCl), with exposure times of 2 h, 2 days, and 7 days, Zhang, Zhang [99] observed induction two hours after the salt treatment of genes related to osmotic regulation (as expected) and TF proteins from several families, including DREB, HD-ZIP, NAC, WRKY, ERF, MYB, bHLH, and AUX/IAA (Supplementary Table S1). These TFs are crucial in regulating gene expression by recognizing *cis*-regulatory elements of the promoters of the genes to be transcribed. The *P5CS1* gene (D-*1*-pyrroline-5-carboxylate synthetase; Supplementary Table S1), related to proline biosynthesis, was identified and was overexpressed after 3 h with 150 mM of NaCl [77].

Zhang, Zhang [99] observed that the *APX4* gene was negatively expressed after seven days of 100 mM of NaCl exposure, which may have reflected the inhibition of ROS accumulation. Despite some differences in methodology (tags versus RNA-Seq), origin of the accessions (Chinese versus Brazilian), exposure times (2 h, 2 days, and 7 days versus 3 h), and stress conditions (NaCl: 100 versus 150 mM), the transcriptomic analyses performed by Zhang, Zhang [99] and Souza, Silva [77] enabled the identification of important convergent genes involved in *J. curcas* responding to the salt stimulus. The data also reinforce the importance of transcriptomic analysis identifying those genes.

The regulation of gene transcription is central to both tissue specific-gene expression and the regulation of gene activity in response to specific stimuli. Thus, whilst some cases of regulation after transcription have been observed, regulation occurs in most cases at the level of transcription via the decision regarding which genes will be transcribed into the primary RNA transcript [74]. In order to produce their effects, transcription factors generally require the ability to bind to DNA and then positively or negatively influence transcription. Our research group recently described the four most outstanding families of transcription factors (ARF, NAC, MYB, and AP2/ERF) that have important roles in responses to the biotic and abiotic stress of plants, especially *J. curcas* [74]. Wu, Xu [104] identified a total of 100 NAC genes (*JcNAC*) in *Jatropha*; based on phylogenetic analysis and gene structures, 83 *JcNAC* genes were classified as members of or separated from 39 previously predicted orthologous groups (OGs) of NAC sequences. *Jatropha* has a single intron-containing NAC gene subfamily that has been lost in many plants. The *JcNAC* genes are non-randomly distributed through the 11 linker groups of the genome of this plant and appear to be preferentially retained duplicates that arose from duplication events. The digital analysis of gene expression has indicated that some of the *JcNAC* genes have tissue-specific expression profiles (for example, in leaves, roots, stem bark, and seeds), and 29 genes respond differentially to abiotic stress.

The MYB family, which is involved in plant development and abiotic stress tolerance, was identified the entire genome of *Jatropha* [105], including in reference to family composition, phylogenetic evolution, and functional prediction analysis. Additionally, they specified the function of the *JcMYB2* gene, which allowed them to identify 128 MYB genes, including 123 *R2R3-MYB*, 4 *R1R2R3-MYB*, and 1 *4R-MYB*.

The AP2/ERF transcription factors comprise the largest family involved in plant hormones such as ABA and ethylene, which help activate ABA and ethylene-dependent and -independent stress-responsive genes. Tang, Liu [106] found that *JcDREB2* was repressed by salinity stress in *Jatropha* leaves. The overexpression of the *JcDREB2* gene in rice reduced tolerance to salt stress, which led to the more pronounced aging and rolling of the leaves; in this study, they reported the low accumulation of proline in the transgenic *JcDREB2*

rice plants compared to the wild-type plants under salinity stress. Likewise, the authors reported higher activities of SOD and CAT in non-transformed rice plants compared to transgenic rice plants under salinity stress, suggesting that they were at a disadvantage in high salinity conditions. Finally, the overexpression of *JcDREB2* was found to improve sensitivity to salt stress in transgenic rice plants. These results have increased our understanding of the roles of this *Jatropha* transcription factor AP2/ERF in plant growth and responses to abiotic stress.

### 3.6. Transgenic J. curcas Plants for Cultivation on Saline Conditions

Improving the salinity and drought tolerance of crop plants by genetic means has been an important but largely unfulfilled aim of modern agricultural development. Though acclaimed widely for its oil, many authors have not considered *J. curcas* economically important enough for domestication, and its seed and oil productivity is hugely variable [107,108]. In terms of viability, a recent study showed that the majority of biofuel projects collapse within the first 5 years of operation [71]. In the case of Ghana, out of the nine verified abandoned *J. curcas* projects for which we could establish a date of collapse, five failed in the first 3 years of operation and the rest failed within the first 5 years. The reasons for these failures were diverse and varied according to projects and location. Ahmed, Campion [71] reported numerous reasons for the failure of *J. curcas* plantations around the world, especially in Ghana, and they found that the main reason for the collapse of *J. curcas* plantations has been the lack of a stable variety, cultivar, or genotype that can guarantee the farmer credibility and stable seed yield of investing in *J. curcas* plantations. However, the affected farmers did not know about this transaction until Biofuel Africa started land clearing to plant *J. curcas*. As a result, there was deep skepticism about the leadership and benevolence of the chieftaincy institutions in the Kusawgu Traditional Are.

*J. curcas* continues to remain semi-wild, and ideal commercial varieties with profitable yields are still lacking. Varieties that produce high and stable yields need to be bred to develop *J. curcas* into a successful biofuel crop [109]. Accordingly, the genetic transformation of *J. curcas* has been pursued for the last 10 years, and many transformation methods have been established [110]. Liu, Wang [111] postulated that overexpressing genes of jasmonate pathway (*JcAOC*) would be valuable in attaining plants with higher tolerance to stresses, including salt stress. These authors reported that the expression of *JcAOC* mRNA was ~2.7-fold higher in the leaves of WT-*J. curcas* subjected to 2 h of 300 mM of NaCl compared to control ones. To confirm whether salt stress and low temperature affected the expression of *JcAOC*, these authors tested the growth performance of *Escherichia coli* cells transformed with recombinant plasmid *pET-JcAOC* under stress conditions and showed that the growth of transformed *E. coli* cells with recombinant plasmid *pET-JcAOC* was 2.5-fold higher than pET-30a empty vectors. Since *E. coli* does not contain the JA biosynthetic pathway, these results revealed a potential property of JcAOC in addition to catalyzing JA biosynthesis, as well as that it is a valuable gene that deserves special attention because the genetic engineering of *JcAOC* could be a potential alternative to enhance *J. curcas* tolerance against salt stress.

Tsuchimoto, Cartagena [112] successfully transformed *J. curcas* into a type that overexpressed the *Synechococcus* GSMT and DMT genes (which encode glycine sarcosine methyltransferase and sarcosine dimethylglycine methyltransferase, respectively, and which catalyze glycine betaine production), and the transgenic plants were noticeably more tolerant to drought than wild-type plants. These findings indicate that these plants can adopt the strategy used by halophytes to survive in drought conditions via the protective nature of glycine betaine [112]. The glycine content was found to be significantly increased in transgenic *J. curcas* plants that overexpressed GSMT and DMT. Jha, Mishra [87] successfully transformed *J. curcas* with the salt-responsive *SbNHX1* gene to enhance its salt tolerance and improve $Na^+$ inside vacuoles, demonstrating the use of this strategy to promote survival under salt stress. An active vacuolar antiporter utilizes the proton motive force generated by vacuolar ATPases and pyrophosphatases to sequester excess $Na^+$ into the vacuole.

Moreover, these authors observed higher Na$^+$ content and less K$^+$ content in transgenic leaf tissues under salt stress, which they attributed to ion homeostasis due to the activity of the vacuolar Na$^+$/H$^+$ antiporter *SbNHX1*. In case, the transgenic lines showed enhanced tolerance at 200 mM of NaCl, enhancing Na$^+$ sequestration into the vacuole.

Salt-tolerant transgenic *J. curcas* plants can be used for cultivation in salt-affected marginal areas for their sustainable management with biofuel production. The diffusion of such transgenic plants may help *J. curcas* during salt cultivation without affecting food production. Recently, Chacuttayapong, Enoki [113] described an efficient method to produce a *Jatropha curcas* transgenic line that expresses *Os03*, *Os04*, *Os08*, *Os10*, and *Os10L*, suggesting that the genes associated with larger seed sizes in *Arabidopsis thaliana* (which were found using the rice FOX-hunting system) produce larger seeds in *Jatropha curcas*.

### 4. Toxicity and Medical Use of *Jatropha curcas* Seeds

The fact that the oil extracted from *J. curcas* is excellent-quality biofuel is indisputable. However, there are currently no agronomically improved varieties available for *J. curcas*, and the seed meal obtained after the extraction of the oil cannot be used as an animal feed due to its toxicity [114]. Nevertheless, in the process of extracting the oil from the seeds of *J. curcas*, a physic nut cake/meal is produced and can be considered a great source of protein and fiber for animal consumption [114,115]. The ability to use *J. curcas* meal as animal feed improves the economics of *J. curcas* production because it means that the crop could produce both fuel and feed [19]. However, the use of this physic nut cake/meal is limited due to the high concentration of two toxic constituents: phorbol esters (PMA) and curcin, as well as other anti-nutritional factors such as lectins, ribosome-inactivating proteins (RIP), and saponins [116–119]. PMA is a diterpene and can act in two ways, acute (intense inflammatory response) and chronic (inducing tumor formation) [120]. Arroyo, Vanegas [121] described a simple, inexpensive method to reduce the PMA to a level of 0.03% using 92% ethanol, a treatment for which no positive test was obtained for tannins and saponins. The ethanol treatment reserves 64% of proteins, 42.7% of soluble carbohydrates, 36.3% of nitrogen sources, and 41.3% of fiber, leaving this methodology as the best option for PMA, saponin, and tannin detoxification. Accordingly, physic nut cake/meal should be approved for addition into livestock feed according to the Food and Agriculture Organization (FAO) criteria [121].

Das, Uppal [122] reported on PMA detoxification through composting using four different animal dung types, and they found that PMA was reduced to 12%, which means that their method was technically less effective than that described by Arroyo, Vanegas [121] using only ethanol.

Some studies have shown the potential of physic nut cake/meal in fish diets. Saha and Ghosh [122] fed rohu (*Labeo rohita*) fingerlings with de-oiled *J. curcas* seed meal (DJSM) fermented for 15 days by an exo-enzyme-producing bacterium, *Bacillus cereus,* isolated from the hindgut of rohu. This procedure caused a decrease in the contents of anti-nutritional factors but increases in the levels of free amino acids and free fatty acids. Hassaan, Goda [123] obtained similar results using DJSM fermented with *Bacillus licheniformis* (LFJSM) and *Bacillus pumilus* (PFJSM). Fermented DJSM was mixed with concentrations of LFJSM and PFJSM, and it was found that PFJSM-25 and PFJSM-50 showed non-significant differences compared to the control and that 50% of fish meal can be replaced by PFJSM in Nile tilapia diets. However, Nile tilapia growth is retarded if they are fed with 0.16% of trypsin inhibitor, as 0.06% is the maximum tolerated level in Nile tilapia growth. A Brazilian study [124] showed that the apparent digestibility coefficients of crude protein for nontoxic and detoxified physic nut cake/meal were 77.51% and 81.11%, respectively—values similar to those reported with another seed meal crude extract. Nontoxic physic and detoxified physic nuts presented apparent digestibility coefficients of 90.48% and 52.10%, respectively [124]. Another Brazilian study [125] demonstrated that 0.025 g of crude extract per kilogram of animal led to 100% animal death, reinforcing the need to detoxify physic nut cake/meal. The acute oral LD50 of the oil was found to be 6 mL kg$^{-1}$ body weight in Haffkine Wistar

strain rats [126]. Elsewhere, the PMA fraction from seed oil was reported as a promising candidate for use as a plant-derived protectant for a variety of crops from a range of pre- and post-harvest insect pests [127]. The results reported so far demonstrate that there is potential for the use of physic nut cake/meal in animal supplementation, mainly in regions not supplied with soybean and its co-products.

Of the toxic compounds present in physic nut cake/meal, curcin is by far the most toxic molecule. Curcin is often classified as a lectin and described as having a similar toxicity to ricin from castor bean. Both curcin and ricin are RIPs that depurinate rRNA, thus arresting protein synthesis. Curcin is a type-I RIP, and ricin is a type-II RIP. Type II RIPs contain a catalytic A-chain (RIP), whereas type I RIPs such as curcin lack this lectin domain. Due to the lack of the lectin domain, the LD50 values of type I RIPs are typically over 1000-fold higher than those observed for type-II RIPs in whole rats [115]. The presence of curcin in the seeds of *J. curcas* is therefore unlikely to present a significant barrier to the processing of *J. curcas* seed meal into animal feed. Additionally, curcin has been investigated for its anti-tumor potential. Some scholars studying curcin have shown that like other RIPs, both purified and recombinant curcin protein can inhibit cell-free translation and protein synthesis in a reticulocyte lysate system [118,128,129]. Consequently, RIPs can arrest DNA and protein metastasis before sequentially inhibiting the division and proliferation of cancer cells [119]. This could be one of the reasons for curcin's anti-tumor activity.Huang, Hou [130] reported that transgenic lines of tobacco plants, expressing *cur2p* fragment (coding premature curcin 2 protein), led to an increased tolerance to tobacco mosaic virus (TMV) and the fungal pathogen *Rhizoctonia solani* by delaying the development of systemic symptoms of TMV and reducing the damage caused by the fungal disease. Zhao, Wang [120] showed the effect of purified curcin from seeds of *J. curcas* on the growth inhibition rate of mouse sarcoma—180 cells; 20% on 3rd day and 40% on 7th day under $100 \text{ mg mL}^{-1}$. The same author also reported that curcin could inhibit S-180 cell growth at $10 \text{ mg mL}^{-1}$. The physic nut cake/meal produced from the non-toxic *J. curcas* variety was used in the preparation of livestock feed at 20, 40, and 60%, replacing soybean meal (which was used as the reference concentrated protein source) [114]. The results of this experiment were encouraging. The animals showed gains in weight and carcass quality similar to those obtained with the soy concentrate, proving the acceptance of the animals. Moreover, Wu, Goh [131] isolated three curcin genes and the endosperm-specific C1 promoter, thus providing useful information and research materials for further functional studies of curcin proteins and genetic engineering of *J. curcas*.

## 5. Concluding Remarks

Salt stress affects various aspects of the biochemistry and physiology of *J. curcas*, with negative impacts on growth and yield, thus supporting the conclusion that the species is sensitive to salt stress. Important genes and proteins involved in a wide variety of processes, including stress signaling, gene expression, redox regulation, defense systems, stress regulation, and secondary metabolism have been studied. Transcriptomic analyses have provided a global overview of genes expressed by the plants after the stress stimulus. Based on transcriptomic studies, differentially expressed genes could be valuable transgenes in transgenic events. Additionally, the development of functional molecular markers based on these differentially expressed genes could be optimized, and a selected panel of these genes could help breeders to select accessions based on gene expression performance. Genetic engineering using halophyte genes might offer an excellent platform for developing glycophytic crops with improved salinity tolerance. It is now possible to comprehend a mechanistic framework of the responses to water deficit and salinity in this still-undomesticated oil seed species. Nevertheless, much work is still required for *J. curcas* to become a real biodiesel feedstock. In summary, the data presented here represent a further step to understand salinity and water-deficit stresses responses in *J. curcas* plants, information that could be used to improve tolerance of this industrial crop. To be environmentally sustainable and non-competitive with existing food sources, the productivity

of *J. curcas* needs to increase to allow for commercial cultivation on marginal land, which would be cost-effective and sustainable only after rational marker-assisted breeding or trait introduction though genetic modification. Taken together, these expression data suggest that the careful selection of candidate genes is a critical step prior to the initiation of a specific genetic engineering project.

Moreover, we have described the beneficial and harmful effects of phorbol ester and curcin from a more global perspective instead of just stating that physic nut cake/meal is a toxic by-product to the environment and animals. This paradigm was overturned with a series of articles describing the effects of both phorbol ester and curcin in *J. curcas* seeds. Furthermore, Embrapa Agroenergia (Brasília, DF, Brazil) recently reported the identification of a non-toxic variety of *J. curcas*. The results achieved so far demonstrate that there is potential for the use of physic nut cake/meal in animal food supplementation, especially in regions not supplied with soybean and its by-products. As a result of the research generated by Embrapa, it is expected in the mid-term to establish viable industrial processes capable of detoxifying *Jatropha* cake/meal and obtaining high-productivity commercial cultivars without the presence of toxic components [132]. Furthermore, Embrapa Semiárido (Petrolina, PE, Brazil) has developed a highly productive variety of *J. curcas* with a yield of 3800 kg of seed per hectare, yielding about 1000 L of oil ha and thus opening the door to a new market in terms of Embrapa biofuels and carbon sequestration, since *J. curcas* is a semi-hardwood plant that can develop good harvests for up to 50 years.

**Supplementary Materials:** The following supporting information can be downloaded at: https://www.mdpi.com/article/10.3390/agriculture12050594/s1, Table S1. Saline stress-associated genes and metabolic pathways in physic nut [133–138].

**Author Contributions:** Conceptualization, M.F.P., A.J.-O. and L.A.R.-P.; data curation, M.F.P., A.J.-O. and L.A.R.-P.; Investigation, M.F.P., A.J.-O. and L.A.R.-P.; methodology, M.F.P., A.J.-O. and L.A.R.-P.; project administration, M.F.P., A.J.-O. and L.A.R.-P.; writing—original draft, M.F.P., A.J.-O. and L.A.R.-P.; writing—review and editing, M.F.P., A.J.-O. and L.A.R.-P. resources, M.F.P. All authors have read and agreed to the published version of the manuscript.

**Funding:** This research was funded by National Council for Scientific and Technological Development (CNPq) (Grants 163524/2017-3).

**Institutional Review Board Statement:** Not applicable.

**Informed Consent Statement:** Not applicable.

**Data Availability Statement:** Not applicable.

**Conflicts of Interest:** The authors declare no conflict of interest.

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
