# Peer review of "Salinity in Jatropha curcas: A Review of Physiological, Biochemical, and Molecular Factors Involved"

_agriculture, doi:10.3390/agriculture12050594_

Round 1

Reviewer 1 Report

Dear Authors

The present manuscript reviewed physiological, biochemical, and molecular factors involved in J. curcas regarding salinity stress. The review includes diverse informations/mechanism involved, well written and presented as well.

Although, Jatropa is known to have "curcin" which has been considered as major limitation for its use. It will be nice to include a small section regarding this issue as well. There are some points in the text which need to be answered, please find them in the manuscript attached herewith.

Thank you

Regards

Author Response

See attachement letter

Reviewer 2 Report

Salt stress affects various aspects of the biochemistry and physiology of J. curcas, with negative impacts on growth and yield, which support the conclusion that the species is sensitive to salt stress. Important genes and proteins involved in a wide variety of processes, including stress signalling, gene expression, redox regulation, defence systems, stress regulation, and secondary metabolism have been studied. The main goal of this review is to compile published results on tolerance / resistance or sensitivity to salt stress in J. curcas. Updating the knowledge on that theme may allow to trace strategies for future research on stress physiology in this promising oil seed species. Overall, the subject of the review is of great interest and worthy. The authors have prepared a nice and attractive review. I suggest some recent reports should also be added to the current review. Other than that, there are a few minor suggestions before publication.

Line 132-151, etc., please check the whole text and define the abbreviations on the first appearance in the main text.

All genes names should be italicized in the text, but not the protein.

Reviewer 3 Report

This manuscript describes the summaries of salt stress study on J. curcas. This manuscript may helpful for future research on stress physiology in oil seed species. There is minor problem that should be corrected in the revised version. 

Line 110-116: Authors should add the references of each of these sentences.

Line 135-150: Authors should add the references of each of these sentences.

All figures are good.

Authors should add a part of prospect of research in J. curcas. Then, please summary the drawbacks of this area.

Finally, this manuscript will be considered to publish in MDPI Agriculture Journal.

Author Response

See letter in attachment
